# Can Wireless Transcutaneous Nerve Stimulation Applied to the Genital Nerve Manage Urinary Incontinence Following Spinal Cord Injury and Multiple Sclerosis?

James Walter [1,2,*], John Wheeler [1,3] and Aasma Khan [4]

1   Department of Urology, Loyola Medical Center, Maywood, IL 60153, USA; jwheele77@gmail.com
2   Research Service, Hines VA Hospital, Hines, IL 60141, USA
3   Department of Surgery, Hines VA Hospital, Hines, IL 60141, USA
4   Department of Psychology, Chicago State University, 60628 IL, USA; cognitive-researcher@outlook.com
*   Correspondence: jameswalter889@gmail.com

**Abstract:** Individuals with spinal cord injury and multiple sclerosis usually use intermittent catheterization for urinary management; however, many patients will also encounter a condition of neurogenic detrusor overactivity, which causes urinary incontinence. The use of muscarinic receptor antagonists is the first-line treatment to manage this condition. These drugs, however, have significant side effects. Transcutaneous electrical nerve stimulation applied to the genital nerve (GEN) is an alternative noninvasive method that produces detrusor inhibition through neuromodulation. Despite studies demonstrating bladder inhibition with GEN, more outcomes are required regarding decreased use of bladder inhibitory medications and concerns with dangling wires. It is proposed that wireless-GEN can be used in home-use studies in order to address these limitations. If needed, wireless tibial nerve stimulation could be added to improve incontinence management.

**Keywords:** lower urinary tract; urinary incontinence; neurogenic detrusor overactivity; catheter; spinal cord injury; multiple sclerosis

## 1. Introduction

Neurological injuries such as multiple sclerosis (MS) and spinal cord injury (SCI) interfere with normal micturition coordinated in the pontine micturition center. These neurological deficits lead to neurogenic detrusor overactivity (NDO), which is nonvolitional bladder contractions that are associated with urinary incontinence. Another pernicious lower urinary tract condition is detrusor sphincter dyssynergia, which produces reflex urethral sphincter contractions during bladder contractions and is associated with high bladder pressures, which are a risk for kidney pathology. Because of these urological conditions, the preferred management method is intermittent catheterization (IC) and using detrusor muscarinic receptor antagonists (MRA) to prevent incontinence between catheterizations [1–4]. The use of MRA is usually effective to prevent incontinence; however, these drugs have some limitations and side effects, including dry mouth, drowsiness, and risks for dementia [1]. Alternative methods to manage incontinence include alternative drug administration methods, botulinum toxin injection into the bladder wall, bladder augmentation, and neuromodulation with electrodes implanted next to the sacral nerves [5–8]. Transcutaneous electrical nerve stimulation applied to the genital nerve (GEN) is an alternative noninvasive but less established method to inhibit NDO. This method relies on activating sensory nerves located in the same spinal dermatome as bladder nerves and inhibits bladder contractions through neuromodulation [8,9]. GEN (dorsal penile nerve in men and clitoral nerve in women) has demonstrated NDO inhibition [8–11]; however, GEN is seldom used in clinical practice. Some limitations have been identified, such as 'dangling wires' and the need for more results about decreased use of bladder inhibitory

medication [9,12–14]. In addition, if GEN were shown to have limitations for managing NDO, further neuromodulation with bilateral tibial nerve stimulation (TIB) could be added to improve incontinence management [14,15].

Fortunately, for the problem of 'dangling wires', a wireless transcutaneous nerve stimulation device, iReliev[R] (model ET-5050, Wireless TENS + EMS Therapeutic Wearable System, ExcelHealth Inc., 5825 Park Vista Circle, Fort Worth, TX, USA) is commercially available. The device consists of stimulating PODs that snap onto the back of surface electrodes (2.75 by 2.75 inches), which are bipolar. The PODs are controlled with a hand-held device, which uses radio frequency communication (Figure 1A). The device can control up to four PODs; however, the device comes with only two PODs (additional PODs can be ordered), and the descriptions below are for use with two PODs. The new device has eight transcutaneous nerve stimulation programs and includes lithium batteries, which are recharged by a USB port. Programs and parameters are selected first, and an ON-OFF button with current adjustment is used for stimulation. The stimulator is a constant-current device, which means that it delivers the same current for different electrode resistances. It has a maximum of 80 mA, divided into 25 equal steps with changes of 3.2 mA per step. One author (Walter) tested the device on his forearm using program #1, consisting of continuous stimulation at 15 Hz and 300-microsecond pulse durations. At step 6 (19 mA), there was a threshold or slight finger movement, which included a slight tingling sensation under the electrode. Moderate movement of the fingers and sensation occurred at steps 7 and 8 (22 mA & 25 mA), and strong finger movement at step 9 (29 mA), which also caused pain under the stimulating electrodes. We recommend that patients experience the device first on their arm, and the identified moderate currents are expected to be used in urinary applications.

**A. Stimulator controller and PODs**

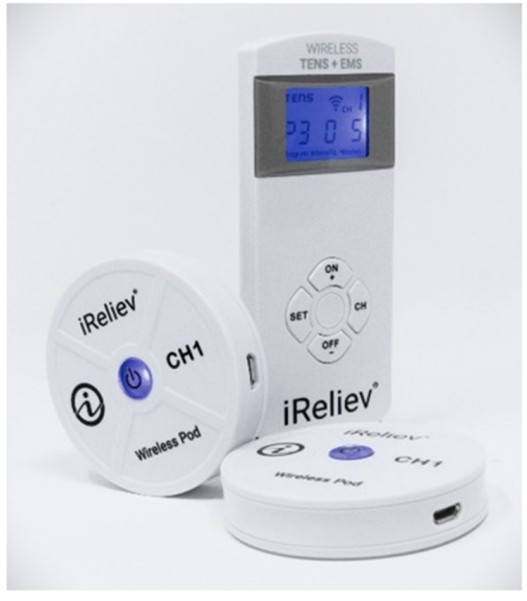

**B. Bipolar surface electrode with POD**

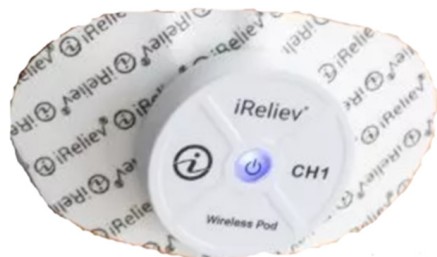

**C. Cut electrode with POD and wire connection**

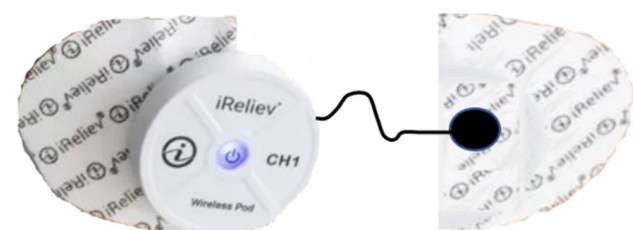

**Figure 1.** Wireless stimulator (iReliev[R]) controller, POD, and surface electrode. (**A**) Controller with two wireless PODs. (**B**) A bipolar surface electrode with POD, which is proposed for unilateral wireless-TIB. (**C**) Two monopolar electrodes from the original electrode that has been cut. This arrangement is proposed for wireless-GEN and bilateral wireless-TIB. The wire and snap connector is shown, see text.

For effective bladder inhibition with wireless-GEN, a moderate stimulation current needs to be used [12,13,15,16]. This current can be ascertained by palpating the anus and identifying moderate anal reflex contractions. Muscle is a proxy for sensory nerve activation

that mediates neuromodulation because only motor responses are easily measured. We recommend program #1, with continuous 15 Hz stimulation, and we expect that current steps 6 or 7 (19 and 22 mA) would be used because the genital nerve lies close to the skin [13]. A moderate current is expected to be obtained without pain and the current should be increased slowly. For stimulation over the shaft of the penis or clitoris, there is not enough room for the POD and the bipolar surface electrode. Thus, the bipolar electrode needs to be cut in half, and two monopolar electrodes should be used (Figure 1C). The POD is then connected to one of the cut surface electrodes and placed on the skin suprapubically. A short connecting wire is then used from the POD to the other half of the surface electrode, which should be placed over the genital nerve. For the male, this electrode should be wrapped around the dorsal side of the shaft of the penis near the pelvis. For females, it should be placed over the clitoris. Some shaving of genital hair is advised for good electrode contact. For Program #1, the 15 Hz stimulation frequency is within the range of frequencies effective for bladder inhibition [8,9,16–20], and the pulse duration is 300 microseconds, which is considered adequate. The program also uses continuous stimulation for up to one hour [12,20].

As introduced above, monoliteral and bilateral wireless-TIB could also be added, if needed, to wireless-GEN to achieve better incontinence management [14,15]. We tested wireless-TIB using one POD applied to the right tibial nerve and the bipolar electrode (Figure 1B). We also tested bilateral tibial nerve stimulation using one POD. The bipolar surface electrode was cut in half One of the cut electrodes was snapped to the POD, and the other was connected to the POD with a wire that ran up and down the pant legs (Figure 1C). The two electrodes were placed bilaterally over the tibial nerve just above the ankle, inside and toward the back of the leg. A current response test was used with program #1, and a moderate stimulation current was observed as moderate toe movements and a moderate tingling sensation under the stimulating electrode at steps 7 and 8 (22 & 25 mA) with further discomfort at the higher step 9 (29 mA), similar to the above responses for the arm. Furthermore, the author observed that a bilateral wireless-TIB with one or two PODs at the same stimulation steps induced the same toe movements and sensation responses, which is what is expected when different size electrodes are used with a constant current stimulator.

Studies involving wireless-GEN must consider the medical conditions of patients with spinal cord injuries and multiple sclerosis. For example, some patients may experience involuntary detrusor contractions, which may cause autonomic dysreflexia. For patients exhibiting these conditions, a blood pressure-lowering medication should be available [12,14]. Furthermore, urinary tract infections require management and most commonly result in incontinence and NDO, which need to be taken into account when evaluating wireless-GEN outcomes [21,22]. Moreover, since wireless-GEN and -TIB are non-authorized treatment options (prescribed off-label), the therapy should only be offered to patients with a low risk for upper urinary tract damage/deterioration evaluated by urodynamics and imaging.

## 2. Future Studies

Testing of wireless-GEN is advisable prior to home use [12]. Cystometry should be conducted with and without stimulation to demonstrate the presence of NDO and the effects of stimulation. Greater bladder capacity during stimulation demonstrates NDO inhibition. If the patient has a recent cystometrogram demonstrating NDO, it may be sufficient to warrant home use of GEN without further tests.

The device training on the arm, described above, should be used. Wireless-GEN methods should be similar for SCI and MS; however, MS patients may have a limitation on the maximal current used to avoid pain [15]. The use of the stimulation should be planned around the patients' symptom goals [12]. Patients with urinary incontinence who are managing it but have a poor tolerance to MRAs may be eligible for this treatment to reduce side effects and reduce their daily doses of medications, and those with MS or incomplete SCI who have partially restored voiding function may be eligible for this treatment to

reduce episodes of urgency and dependence on diapers. These various goals will inform the start of wireless-GEN use.

Frequent conversations should be conducted with the patient at the start of GEN, and it is essential to document any difficulties and urological experiences [12]. For the first two days to a week of wireless-GEN, it is suggested to continue the usual individual dosage of MRA (if used). The time to start stimulation before the subsequent catheterization should be adjusted, as needed, based on urinary incontinence risk. The surface electrodes can remain on the patient for 24 h but should be checked daily for skin irritation and replaced as needed. The surface electrodes can be cleaned with water and reused. If the skin becomes irritated, the electrodes can be removed during non-stimulation periods. If there is a need not to use the GEN electrodes for more extended periods, then TIB could be used.

If incontinence continues to be prevented during the start of GEN use, the bladder medication should be reduced by 50%. If continence is maintained for the next two days to a week of stimulation, the medications should be stopped. If incontinence returns, the medication needs to be restored. A higher stimulation current, a lower stimulating frequency, or adding wireless-TIB could be considered if poor continence assistance is provided with wireless-GEN [21]. If continence can be maintained with wireless-GEN alone, then it would be essential to demonstrate in the following few weeks if less stimulation is also effective. This testing could include shorter stimulation periods and lower stimulating current. Following these tests, wireless-GEN could continue unchanged. Two or more months of home use of wireless-GEN would be expected to establish effectiveness [12,21]. The patients should be aware that they can stop wireless-GEN at any time if they feel it is not helping them to achieve their urinary goals or if they prefer their prior drug therapy for incontinence management.

Nighttime use of GEN will be more problematic for incontinence management because of the long period between catheterizations [12,13,16]. Patients may be advised to delay nightly use of wireless-GEN until the effectiveness of daytime use is established in order to avoid interfering with their sleep. During the night, males often use external condom-collection devices, and females will often use pads; their use and amounts of incontinence should be recorded. Suppose the one-hour limitation of the current wireless stimulator for the GEN device is insufficient to manage nighttime incontinence. In that case, a wired stimulator that has unlimited stimulation time should be considered.

This type of wireless stimulator can achieve other therapeutic effects beyond the management of neurogenic bladder problems. For example, for the treatment of muscle pain, transcutaneous nerve stimulation can be applied over the area of discomfort and at a moderate current [23]. The device also offers neuromuscular stimulation programs for applications such as disuse atrophy and spasticity. Decreased spasticity and increased muscle girth have been reported [24]. The leg extensor and flexor muscles are stimulated simultaneously for isometric muscle contractions and with an ON-and-OFF program. It would be beneficial to have patients use the stimulation for these applications when not using it for neurogenic bladder concerns.

Following a study as outlined above, analysis of results would need to include patient baseline characteristics such as the type and time since injury for SCI, the severity of MS by the expanded disability status scale, as well as the patient's gender, the underlying neurogenic lower urinary dysfunctions, and if there are also bowel symptoms or sexual impairments. Results should be grouped by the patient's urinary goals and wireless-GEN methods. A voiding diary and/or intermittent catheterization could be used to determine bladder capacity and urinary incontinence episodes, as well as whether NDO is present. Patients' perceptions and satisfaction of wireless-GEN or any adverse events are also essential to report. In order to advance patient care, it would be beneficial to report the benefits and harms of wireless-GEN and the concomitant stimulation of the neuromuscular or tibial nerves.

**Author Contributions:** Concept, writing, review and editing—J.W. (James Walter), J.W. (John Wheeler) and A.K. All authors have read and agreed to the published version of the manuscript.

**Funding:** This research received no external funding.

**Institutional Review Board Statement:** Not applicable.

**Informed Consent Statement:** Not applicable.

**Data Availability Statement:** Not applicable.

**Acknowledgments:** We thank Achim Herms, Innsbruck Austria, and Stefania Musco, Florence Italy, for their guidance in this submission. The current authors are members of an International Neuro-Urology Research Group, which is supporting this report and related work in this area.

**Conflicts of Interest:** The authors declare no conflict of interest.

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
