# Peer review of "Can Wireless Transcutaneous Nerve Stimulation Applied to the Genital Nerve Manage Urinary Incontinence Following Spinal Cord Injury and Multiple Sclerosis?"

_2673-4397, doi:10.3390/uro2030021_

Round 1

Reviewer 1 Report

This article proposes an idea of the application of a wireless and wearable device for the intervention of bladder over-activity. However, some issues need major revisions.

Some keywords authors wrote were not presented in Abstract. Please check again and revise this problem.

Line 19. “bladder inhibition” should be “the inhibitory effect on micturition”. Authors should emphasize the inhibitory effects of electrical nerve stimulation on voiding.

Line 110, 139. “bladder inhibition” should be “the inhibition of micturition”. The emphasis should be specific, as reviewer mentioned above.

Line 37 to 40. Authors could mention more therapeutic approaches or the difficulties for treating bladder disorder. Authors might refer to recent articles, as follow: (https://pubmed.ncbi.nlm.nih.gov/33925860/) (https://pubmed.ncbi.nlm.nih.gov/32500759/).

Figure 1B could be drawn with more delicate manner or presented with the actual photo. The graphic illustration of application on human body or manikin could be added.

Line 98. Authors could explain “the lower current steps, 6 or 7”. Readers must be informed with some specifications of product (iRelievR). A table includes simple descriptions of the steps and the programs could be added.

Line 126. “Additional medical issues related to SCI and MS urological care…” is odd. Do authors meanAdditional medical issues other than SCI and MS urological care…”?

Line 171. “Stimulation holidays should also be considered to demonstrate that continued GEN is needed.” Is kind of confusing. Please add the descriptions about the typical schedule of stimulation.

Line 195. “It would be beneficial to have patients use the stimulation device for these applications when not using it for neurogenic bladder problems.” could be amended. Reviewer hypothesize that authors mean “This type of wireless device can achieve other therapeutic effects except for the management of neurogenic bladder problems.”

Line 201. “Patients’ perceptions of GEN are also essential to report. Did they have any problems with GEN, are they satisfied, and would they recommend it to others?” is too colloquial. Please follow the manner of academic writing.

Line 207. In the section of Acknowledgement, just mention who provide the assistance or the guidance. You don’t need to write the goal of your team or why you advocate this issue. The hope of this publication should only be mentioned in the cover letter, if any, during submission. The information about whom to contact should only be mentioned in the cover letter, too. Please refer to the previous published papers, and revise this section, or check the submission guidelines again.

Some descriptions could be more specific and written with appropriate words or according to scientific evidence. Examples are as follows:

Line 16. “this condition” should be “NDO”.

Line 16. “adequate” for attenuating symptoms.

Line 17 and line 36. “significant side effects” is inappropriate. These anti-cholinergic drugs usually work, whereas they have some limitations and some inevitable side effects, but not “significant”.

Line 20. “the need for more outcomes” could be more specific. What kind of outcomes are those?

Line 23. “with the goal of improved NDO management” should bewith the goal of improving NDO management”.

Line 35. “Bladder inhibition to manage incontinence frequently involves anticholinergic medication” could be revised. One of the current preferences for managing incontinence is the inhibition of micturition through anticholinergic medication.

Line 41. “an alternative noninvasive but less established method” could be “an alternative noninvasive method, which is not fully established”.

Line 47. What is “the need for more reports”? It needs more detailed descriptions.

Line 50. What kind of outcomes are these?

Line 71. “urinary applications” could be “the applications for alleviating detrusor overactivity”.

Line 151. Do authors mean “These various goals will facilitate/promote/advocate the start of GEN use.”?

Line 199. “urinary goals” could be more specific. Perhaps “the goals for the management of NDO”

A few writing mistakes should be amended. As below:

Line14. There is an extra hyphen. “have-neurogenic”

Line 41. What is “eh genital nerve”?

Line 103. “The POD, therefore, should…” should be “Therefore, the POD should…”

Line 136. “Although not required…” could be “Although it is not necessarily required…”. Please be careful to the descriptions relating clinical application. Some descriptions may lead to the misuse of these self-care devices.

Line 162. “Suppose incontinence continues to be prevented after the first two days of GEN.” Please check the grammar.

Author Response

attached

I hope revised paper is also downloaded

Reviewer 2 Report

This paper describes the application of wireless transcutaneous nerve stimulation to the genital nerve for urinary incontinence management in neurogenic bladders following spinal cord injury and MS. The background is described as such that while anticholinergic medication is effective, it has significant side effects. Alternative methods, including transcutaneous nerve stimulation, are presented to circumvent these side effects. Stimulation of the genital nerve has demonstrated neurogenic detrusor overactivity with some limitations. Dangling wires are one of these limitations, which can be avoided using the iReliev-POD. The device is then described, including respective programs and settings. 

The manuscript is nicely written and informative but slightly too descriptive. I have not much to add to it except for the following:

- regarding the anticholinergic side effects, what percentage of patients are affected by these in the patient cohort?

- How many need an alternative treatment?

- Why is GEN seldom used in clinical practice? What this implies is what is the magnitude of the patient population actually in need of this device? 

Author Response

attached 

I hope revised manuscript is also loaded

Round 2

Reviewer 1 Report

The manuscript was thoroughly amended.

In addition, a few mistakes or problems can be revised. The manuscript can be accepted after minor revision.

Problems and suggestions are as follows.

In line 14. many “patients” will also

In line 17. these drugs have significant also cause several side effects.

In line 22. It is proposed that wireless-GEN “can” be used with “in” home-use studies in order to address these limitations.

In line 127. “Furthermore”, the author observed that…

In line 154 and 162. What is “MRAs”? Is it muscarinic receptor antagonists?

In line 208-209. “as well as whether NDO” is kind of confusing. Would you mind check again?

Author Response

Reviewer 1, second round of comments:

The manuscript was thoroughly amended. In addition, a few mistakes or problems can be revised. The manuscript can be accepted after minor revision.

Response: we appreciate the additional comments and have revised as suggested.

Problems and suggestions are as follows.

In line 14. many “patients” will also; Response: revised

In line 17. these drugs have significant also cause several side effects. Response: we could not follow your suggestion; made the following revision: These drugs, however, have significant side effects.

In line 22. It is proposed that wireless-GEN “can” be used with “in” home-use studies in order to address these limitations. Response: thank you, revised

In line 127. “Furthermore”, the author observed that… Response: revised as suggested.

In line 154 and 162. What is “MRAs”? Is it muscarinic receptor antagonists?

Response: we revised with defining MRA this earlier in the paper:

Because of these urological conditions, the preferred management method is intermittent catheterization (IC) and using detrusor muscarinic receptor antagonist (MRA) to prevent incontinence between catheterizations [1-4]. The use of MRA is usually effective to prevent incontinence; however, these drugs have some limitations and side effects, including dry mouth, drowsiness, and risks for dementia [1].

And then refer to it at the end of the paper on lines 154 and 162

Patients with urinary incontinence who are managing it but have a poor tolerance to MRAs may be eligible for this treatment to reduce side effects and reduce their daily doses of medications; and those with MS or incomplete SCI who have partially restored voiding function may be eligible for this treatment to reduce episodes of urgency and dependence on diapers.

For the first two days to a week of wireless-GEN, it is suggested to continue the MRAs usual individual dosage (if used).

In line 208-209. “as well as whether NDO” is kind of confusing. Would you mind check again? Response: revised: A voiding diary and/or intermittent catheterization could be used to determine bladder capacity and urinary incontinence episodes, as well as whether NDO is present.
